# Application of Ultrafiltration in a Swimming Pool Water Treatment System

**DOI:** 10.3390/membranes9030044

**Published:** 2019-03-25

**Authors:** Mariusz Dudziak, Joanna Wyczarska-Kokot, Edyta Łaskawiec, Agnieszka Stolarczyk

**Affiliations:** 1Institute of Water and Wastewater Engineering, Faculty of Energy and Environmental Engineering, Silesian University of Technology, Konarskiego 18, 44-100 Gliwice, Poland; mariusz.dudziak@polsl.pl (M.D.); edyta.laskawiec@polsl.pl (E.Ł.); 2Department of Physical Chemistry and Technology of Polymers, Faculty of Chemistry, Silesian University of Technology, M. Strzody 9, 44-100 Gliwice, Poland; agnieszka.stolarczyk@polsl.pl

**Keywords:** swimming pool water, water treatment, ultrafiltration, membrane cleaning

## Abstract

Swimming pool water was treated using an ultrafiltration process using ceramic and polymer membranes for comparison. It was determined that the efficiency of the process depended on the type of membrane used. The polymer membrane decreased the absorbance and concentration of combined chlorine in the pool water to a greater extent than the ceramic membrane. In the case of a ceramic membrane, the concentration of combined chlorine in the permeate exceeded the limit values. During the ultrafiltration process, the permeate flux decreased, causing the blockage of membrane pores. The extent of this phenomenon was similar for both tested membranes. In the case of the ceramic membrane, flushing it with water could significantly restore its initial performance. For both tested membranes, a high regeneration efficiency was observed during chemical treatment with an alkaline solution. SEM photos of the polymer membrane showed low resistance of this polymer to the chlorine present in the swimming pool water.

## 1. Introduction

Swimming pool users introduce fragments of the epidermis, sweat, urine, and microorganisms into the water, including pathogenic microorganisms which thrive in such an environment. Therefore, swimming pool water is a mixture of water added to a closed circulation system (usually from the mains) and water from a swimming pool that is constantly being treated and disinfected.

The main factors affecting the quality of water in the pool, ensuring health and safety and the comfort of bathers, are primarily modern technologies for water treatment, with a high efficiency of disinfection and filtering and the selection of appropriate chemicals. These factors guarantee that the pool water meets strict requirements in terms of physicochemical and bacteriological parameters [1,2,3].

Due to the fact that disinfection byproducts (DBP) were proven to pose a threat to bathers’ health, it is now standard practice to support conventional treatment methods (filtration, usually preceded by coagulant dosing + disinfection with chlorine compounds) with processes that boost the chlorination process, for instance, irradiating the flux of water with ultraviolet (UV) light or ozonization. Equally effective in the area of pool water treatment technology, other methods include hydrogen peroxide biocides, mixtures of chlorine and chlorine dioxide, titanium dioxide, silver nanoparticles, and heterogeneous catalysts. Introducing these solutions to swimming pool water treatment systems allows reducing the amount of chlorine that is added at the final stage of the disinfection process. This, in turn, allows lowering the concentration of chloro-nitrogen compounds that cause allergies and irritation of the upper respiratory and digestive systems, and those that are also mutagenic and responsible for the characteristic unpleasant smell of swimming pool water [4,5,6,7,8].

Alternatively to swimming pool water treatment technologies, one can also use hybrid systems that combine filtration with advanced oxidation processes [9,10].

Currently, another method that is considered to be an effective swimming pool technology is membrane water treatment, including specifically the ultrafiltration process. Ultrafiltration captures fine solids, colloids, bacteria, and viruses. The transport mechanism is a sieve-like structure which does not allow solids larger than the pore diameter to pass through. The process transmembrane pressures fall within the range of 0.1–1.0 MPa. These membranes are made of ceramic and polymer materials [11,12]. Considering the properties of ultrafiltration in swimming pool water treatment technology, this process could be applied in place of classical filtration performed on a sand–anthracite bed.

The main disadvantages of the membrane processes are the lowered hydraulic performance of the membrane caused by adverse effects of the filtration process (i.e., concentration polarization), development of a gel layer on the surface of the membrane, accumulation of dirt on the surface of the membrane or inside its pores (fouling), and precipitation of salts poorly soluble in water that create inorganic sediments (scaling) [13]. These phenomena occur concurrently and their adverse effects add up. As a result, the permeate flux is continually decreasing during the membrane system’s operation. The problems mentioned above make it necessary to periodically clean the membranes during their operation [14]. The conditions for cleaning the membranes are usually chosen by trial and error.

The water that is being treated may contain compounds that are damaging to the membrane, such as chlorine [15]. Because of that, the membrane’s total operation time is decreased.

During in-house research, two ultrafiltration membrane types were selected for comparison and used to support the sports swimming pool water treatment system in a chosen indoor facility. The effectiveness of the pool water treatment, the intensity of the adverse effects, and the possibility of cleaning the membranes were assessed.

## 2. Materials and Methods

### 2.1. Research Subject

The subject of research involved water samples taken from a sports and leisure swimming pool basin, using a vertical water flow system (capacity: 188.6 m^3^/h, filtering speed: 30 m/h). The dimensions of the swimming pool basin (25 m × 12.5 m × (1.1–1.8) m) allow for approximately 30 persons to swim comfortably. Water deficits that are the result of evaporation, splashing, and the need to flush filter beds are replenished by drawing water from the mains into the retention tank. A closed circulation system with an active overflow is the basic requirement for correct circulation of swimming pool water (Figure 1). The treated water is introduced to the swimming pool through two DN160 channels located at the bottom of the pool basin. The water is received by two overflow troughs placed alongside the swimming pool walls and directed to the retention tank through six outlets. The next stage involves suction using the circulation pump equipped with a basket and a mesh filter (the so-called prefilter), where large solid contaminants are captured. The pump moves the water to two DN2000 filters where, after the application of a pH correction solution (50% solution of H_2_SO_4_), UV irradiation (low-pressure UV lamp, 1.8 kW), and final disinfection (14% solution of NaOCl), it is directed to the swimming pool through heat exchangers. Before the filters, a coagulant solution is applied (5% solution of aluminum hydroxychloride). The swimming pool water treatment system has two pressure filters with a 1.2-m-high multi-layered sand–anthracite filter bed.

### 2.2. Assessment of Swimming Pool Water Treatment Process

The effectiveness of the pool water treatment process was determined on the basis of physicochemical analyses of the feed material samples, permeate (taken at specific intervals), and retentate. The analyzed parameters of swimming pool water quality were pH, turbidity, color, absorbance of UV_254_, free chlorine, total chlorine, combined chlorine, ammonia nitrogen, and nitrate nitrogen. The difference between the concentration of total and free chlorine allowed determining the concentration of combined chlorine in tested samples.

The general parameters were measured with an inoLab^®^ 740 laboratory multi-parameter meter manufactured by WTW. The absorbance was measured at 254 nm, using a UV–visible light (UV–Vis) Cecil 1000 (Analytik Jena AG company, Jena, Germany). To determine the turbidity of samples, a Turbidimeter TN-100 (EUTECH, Singapore) was used. Color measurements were performed with a UV–Vis Spectroquant^®^ Pharo 300 (Merck, Kenilworth, NJ, USA) at 340 nm. Concentrations of total and free chlorine were determined with a colorimetric method using a portable Pocket Colorimeter II Device^TM^ (Hach^®^, Loveland, CO, USA). The concentrations of ammonia nitrogen and nitrate nitrogen were determined with a Photolyser 400 tester (Dinotec, Sevilla, Spain).

Table 1 presents the results of the basic analysis of the swimming pool water quality.

### 2.3. Membrane Filtration

The membrane filtration was conducted in two circuits operating in a cross-flow system using pipe membranes made of different materials (Table 2). The first filtration circuit was entirely made of steel and equipped with a pipe module adjusted to polymer membranes with an active surface of 240 cm^2^, an intermediate tank with a volume of 15 dm^3^, a high-pressure pump with a capacity between 0.5 and 3.0 m^3^/h (type CRN 3) produced by Grundfos, and a control and measurement apparatus. The picture of the first filtration system is shown in Figure 2.

The second filtration circuit had a structure similar to the first, but this one was adjusted to have ceramic membranes. This circuit had a low-pressure pump of capacity between 1.50 and 3.50 m^3^/h (type CRN 1) produced by Grundfos.

Membrane filtration tests were carried out in the following order: membrane conditioning with deionized water, pool water filtration, and an attempt to clean the membranes. The process was conducted at 0.1 MPa transmembrane pressure conditions for the ceramic membrane, and at 0.2 MPa for the polymer membrane. These pressures were selected on the basis of preliminary tests.

The temperature of the filtered water was kept at a constant level of 20 ± 2 °C. The filtration process was carried out to the maximum degree of feed volume reduction ratio (VRR, %), which did not cause hydraulic disturbances in the system. This was determined visually based on the evaluation of the water table in the feed tank.

The process of cleaning the membranes was a multistage one and consisted of flushing them with deionized water (cycle 1), and chemical cleaning with a 0.3% solution of HNO_3_ acid (cycle 2) and a 1% solution of NaOH (cycle 3).

To determine the effectiveness of the membrane process, it was necessary to establish its performance by calculating the permeate flux volume (Equation (1)) and the retention/reduction (Equation (2)) of the chosen pollutants or the general indicators associated with the presence of particular groups of pollutants.
(1)$$J_{v}\left(J_{w}\right)=\frac{V}{F\cdot t},$$
where *J_v_* (*J_w_*) is the volumetric permeate flux (m^3^/m^2^∙s) for swimming pool water (for deionized water), *V* is the volume (m^3^), *F* is the membrane area (m^2^), and *t* is the time (s).
(2)$$R=\;\left(1-\frac{C_{p}}{C_{f}}\right)\cdot 100\%,$$
where *R* is the retention (%), *C_p_* is the concentration in permeate (mg/dm^3^), and *C_f_* is the concentration in feed (mg/dm^3^).

The phenomenon of blocking the membrane surface was determined by calculating its relative permeability (Equation (3)), with a quotient of fluxes (average value) specified for a new membrane during its conditioning with deionized water (*J_w_*) and after the pool water treatment (*J_v_*).
(3)$$\alpha =\frac{J_{v}}{J_{w}}.\;$$

The effectiveness of membrane cleaning was determined by calculating the so-called recovery ratio (Equation (4)).
(4)$$MPR=\left(\frac{J_{v\left(cleaned\right)}-J_{v\left(fouled\right)}}{J_{w}-J_{v\left(fouled\right)}}\right)\cdot 100\%,$$
where *J_v(cleaned)_* is the volumetric permeate flux for the cleaned membrane (m^3^/m^2^∙s), and *J_v(fouled)_* is the volumetric permeate flux for the fouled membrane (m^3^/m^2^).

The stability and degree of contamination were evaluated via imaging with the SEM Phenom proX microscope (Phenom-World, Waltham, MA, USA).

## 3. Results and Discussion

The initial hydraulic value of the polymer membrane was almost five times lower than that of the ceramic membrane (Figure 3). The average permeate flux volume of deionized water of the ceramic membrane was 75.9 × 10^−6^ m^3^/m^2^·s, and that of the polymer membrane was 17.0 × 10^−6^ m^3^/m^2^·s. This is mainly due to the characteristics of the used membranes, including the materials (contact angle) they were made of (Table 2). The operation of both membranes was very stable and their performance did not vary considerably during the filtration process. This experiment was repeated four times, and the observed differences in volumetric permeate flux values did not exceed 5%.

During the swimming pool water treatment, the hydraulic performance of both the polymer and ceramic membrane decreased with filtration time (Figure 4). This was caused by the blockage of membrane pores with contaminants present in the water. Because of that, the average values of relative permeate flux α of the tested membranes were compared. Parameter α is an intensity measure of the phenomenon of membrane pore blocking. It was determined that its value was similar for both membranes. The degree of feed volume reduction ratio (VRR) for the polymeric membrane was 34.8%, and that for the ceramic one was 51.2% (Table 3).

The analysis of SEM micrographs of the polymer membrane showed not only the contamination of its surface with sediments (Figure 5), but also the damage done to the polymer structure due to chlorine compounds present in the swimming pool water (Table 1). This indicates the low resistance of this polymer to the chlorine present in the swimming pool water. Over time, this phenomenon may also reduce the effectiveness of the membrane in the elimination of pollutants from the pool water.

Next, the effectiveness of eliminating particular contaminants (i.e., the general indicators associated with the presence of particular groups of contaminants) was evaluated. The parameters of permeate and swimming pool water quality were compared against limit values specified in World Health Organization (WHO) guidelines [1], DIN 19643 [2], and the ordinance of the Polish Minister of Health [3].

The ultrafiltration process completely removed the water color. The elimination of other contaminants was dependent upon both the membrane type and the filtration time (Figure 6). The polymer membrane achieved better results in terms of lowered absorbance and combined chlorine concentration than the ceramic membrane.

On the other hand, the ceramic membrane eliminated water turbidity to a greater extent than the polymer one. The effectiveness of lowering this ratio by the ceramic membrane increased with filtration time. This was caused by the condensation conditions that allowed for better separation of the contaminants. On the other hand, the concentration of nitrogen compounds in the permeate, including primarily the ammonium nitrogen, was close to or higher than the values specified for the water for treatment. A higher concentration of ammonium nitrogen in the permeate may indicate the occurrence of additional reactions during the water treatment process, including releasing this contaminant from complex compounds. The mechanisms for removing contaminants during the ultrafiltration process were presented in Reference [16]. In the retentate flux, the majority of contaminants became concentrated.

It was determined that flushing the ceramic membrane with water (cycle 1, Figure 7) may significantly restore its initial performance. This proves that the adverse effect of blocking the membrane surface is partially reversible.

The chemical cleaning of membranes with an alkaline solution, performed at a later stage, caused a significant increase in the initial performance of both tested membranes (cycle 3). The polymer membrane was regenerated to 98% and the ceramic membrane to almost 100% (Figure 7). The high regeneration effectiveness of the polymer membrane was also confirmed by SEM micrographs of its surface (Figure 8). The results of cleaning the membranes with an acid solution (cycle 2) were similar to the ones obtained by flushing them with deionized water (cycle 1).

Cleaning membranes with chemicals gives, in the majority of instances, better results than flushing them with water, and it increases the chance of restoring the membranes’ initial performance [17]. It is the most significant stage in the entire process of membrane cleaning. The type of cleaning agent depends on the character of the sediments that are deposited on the surface of the membranes [18,19]. In the case of inorganic compounds, including salt crystals, solutions of acids were found to be effective. Organic compounds can be dissolved in alkaline solutions. Comparing the above results to the ones obtained as part of the research allows concluding that the swimming pool water under treatment was contaminated to a more significant extent with organic substances than with inorganic ones. Among the organic contaminants found in the swimming pool water [20], we can find urea, amino acids, creatinine, trihalomethanes, haloacetic acids, halogenoacetonitriles, chloramines, and nitrosamines. Swimming pool water may also contain residuals of various cosmetics used by bathers.

## 4. Conclusions

Studies on the use of ultrafiltration in swimming pool water treatment systems showed some limitations in the use of this process. This research demonstrated that the effectiveness of swimming pool water treatment is significantly influenced by both the type of the membrane and the time of filtration. The polymer membrane decreased the absorbance and concentration of bound chlorine in the swimming pool water to a greater extent than the ceramic membrane. In the case of the ceramic membrane, the concentration of bound chlorine in the permeate exceeded the limit values. Chlorine present in the swimming pool water damaged the polymer membrane structure, which may decrease its total operation time. Over time, this phenomenon may also reduce the effectiveness of the membrane in the elimination of pollutants from the swimming pool water. During the ultrafiltration process of the swimming pool water, the permeate flux decreased due to the blocked membrane pores. The extent of this phenomenon was similar for both tested membranes. In the case of the ceramic membrane, this adverse effect was partially reversible. For both tested membranes, a high regeneration efficiency was observed following chemical treatment with an alkaline solution. This means that the swimming pool water under treatment was contaminated to a more significant extent with organic substances than with inorganic ones.

## Figures and Tables

**Figure 1 membranes-09-00044-f001:**
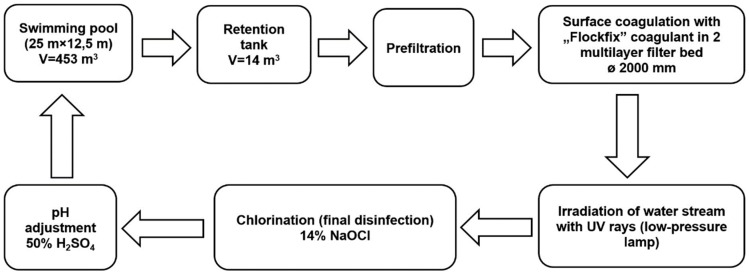
The scheme of water treatment in tested swimming pool.

**Figure 2 membranes-09-00044-f002:**
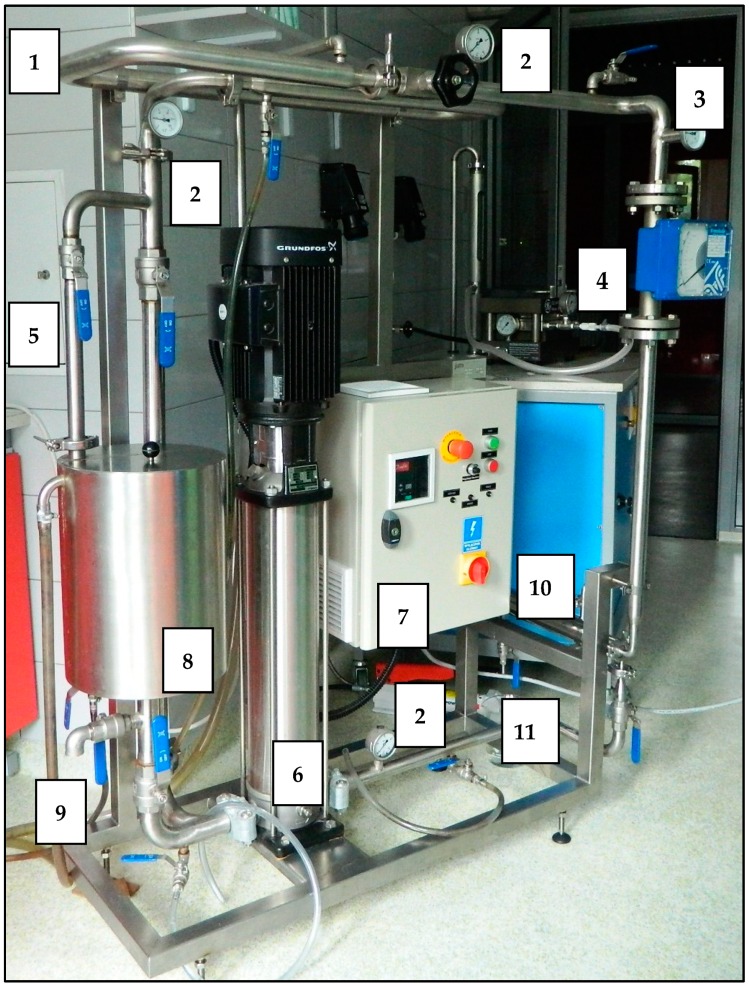
Picture of the first filtration system (1—water coat, 2—manometer, 3—thermometer, 4—flowmeter, 5—regulating valves, 6—high-pressure pump, 7—control box with inverter, 8—tank, 9—drain from the tank, 10—membrane module, 11—permeate outlet).

**Figure 3 membranes-09-00044-f003:**
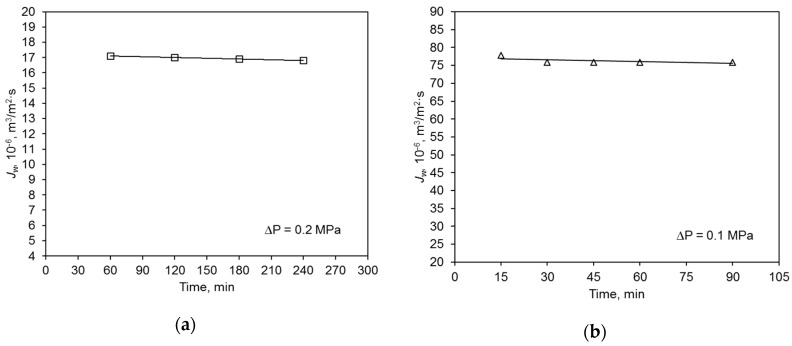
Hydraulic membrane capacity for deionized water: (**a**) polymer membrane, (**b**) ceramic membrane.

**Figure 4 membranes-09-00044-f004:**
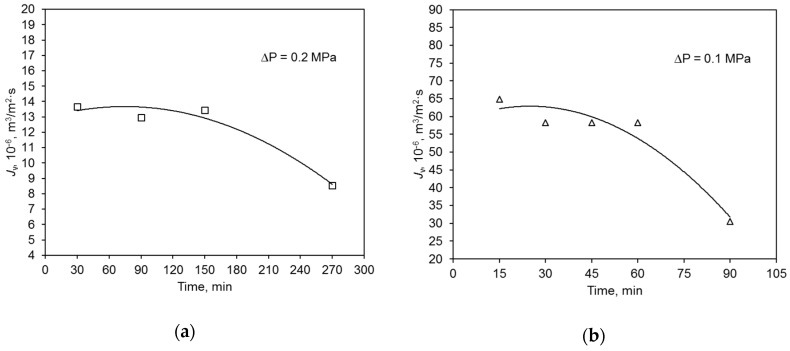
Change of hydraulic membrane capacity during swimming pool water filtration: (**a**) polymer membrane, (**b**) ceramic membrane.

**Figure 5 membranes-09-00044-f005:**
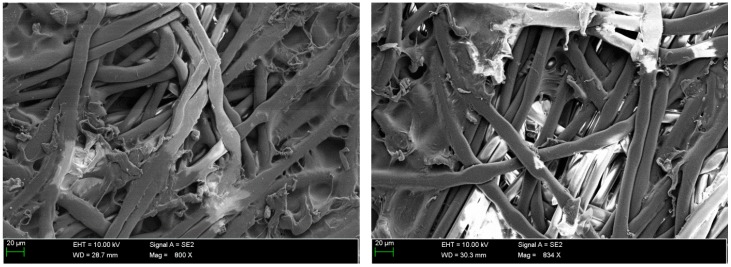
SEM micrographs of the membrane after filtration of swimming pool water.

**Figure 6 membranes-09-00044-f006:**
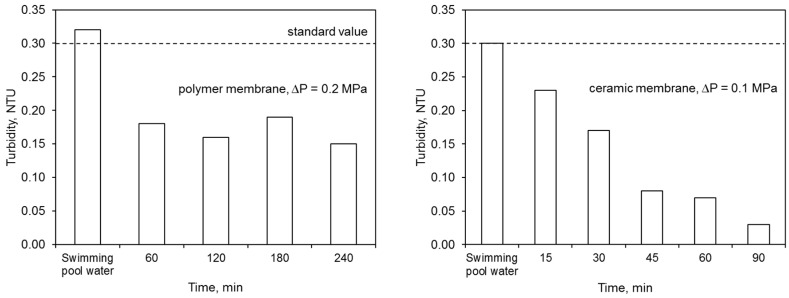
Efficiency of ultrafiltration: (**a**) polymer membrane, (**b**) ceramic membrane.

**Figure 7 membranes-09-00044-f007:**
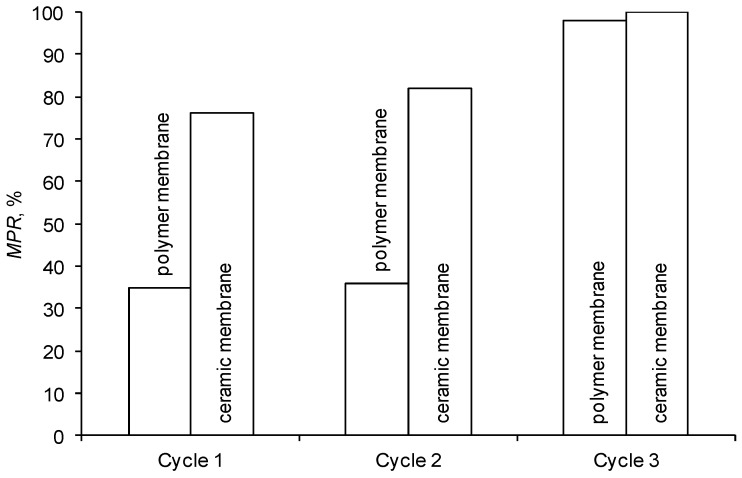
Efficiency of membrane cleaning.

**Figure 8 membranes-09-00044-f008:**
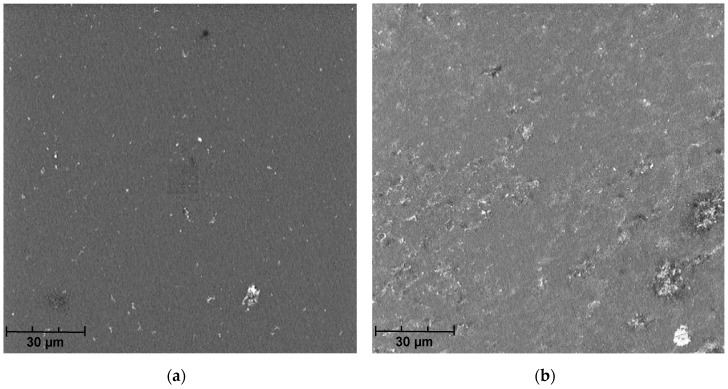
SEM micrographs from (**a**) the polymer membrane prior to use, (**b**) after filtration of swimming pool water, and (**c**) after chemical cleaning.

**Table 1 membranes-09-00044-t001:** The results of basic quality analysis of tested swimming pool water.

Parameters	Unit	Value
pH	-	7.74
Turbidity	NTU	0.30
Color	mgPt/L	2.0
Absorbance	1 cm	0.045
Free chlorine	mgCl_2_/L	0.33
Total chlorine	mgCl_2_/L	0.80
Combined chlorine	mgCl_2_/L	0.47
Ammonium nitrogen	mgN-NH_4_/L	0.13
Nitrate nitrogen	mgN-NO_3_/L	5.00

**Table 2 membranes-09-00044-t002:** Characteristics of the membranes (manufacturer data).

Membrane Type	ES625	-
Manufacturer	PCI Membrane System Inc.	TAMI Industries
Membrane material	polyethersulfone	TiO_2_
Max. temperature, °C	80	150
Max. pressure, MPa	1.5	9.0
pH range	1.5–12	0–14
Molecular weight cut-off, kDa	25	8
Contact angle ^1^, °	60	41
Membrane area, m^2^	0.024	0.350

^1^ determined experimentally using a PG-1 goniometer from Fibro System AB.

**Table 3 membranes-09-00044-t003:** Transport properties of membranes during swimming pool water treatment. VRR—volume reduction ratio.

Parameter	Polymer Membrane	Ceramic Membrane
*J_v_*, ×10^−6^ m^3^/m^2^·s	12.1	54.1
VRR, %	34.8	51.2
α, -	0.72	0.71

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
