# Peer review of "Application of Ultrafiltration in a Swimming Pool Water Treatment System"

_membranes, 2019, doi:10.3390/membranes9030044_

Round 1

Reviewer 1 Report

This manuscript reports the comparison of ceramic and polymeric membranes in the treatment of swimming pool water via ultrafiltration. Also, the authors have assessed the effect of chlorine present in the swimming pool water on the polymer membrane stability. I would recommend the acceptance of this manuscript after few minor revision as mentioned below.

1.      Abstract – In the sentence ‘Better results were obtained with the use of polymer membrane’, what does ‘better results’ denote? Better results in terms of? From the results, it seems that ceramic membranes are working better.

2.      Pg 2 Line 60 – Check the sentence ‘The water flux…..’. It has to be water, not water flux.

3.      Pg 2 Line 51 – the disadvantage of using membranes to treat pool water has been mentioned. But in the next paragraphs, the novelty of this work has to be explained. Also, how does the current work overcome the disadvantage may have a good gist of what the paper is all about.

4.      Section 2.3 – Can the authors give a schematic to better understand the circuits involved in membrane filtration using ceramics and polymers?

5.      Pg 4 Line 118 – Why have the authors chose different working pressures (0.1 MPa for ceramics and 0.2 MPa for polymer membranes)? In that case will the comparison on flux and fouling be reasonable?

Author Response

Response to the review comments

 Thank you for all the comments. The work has been improved.

 Review Report to Reviewer 1

 1.      Abstract – In the sentence ‘Better results were obtained with the use of polymer membrane’, what does ‘better results’ denote? Better results in terms of? From the results, it seems that ceramic membranes are working better.

 It has been improved.

 2.      Pg 2 Line 60 – Check the sentence ‘The water flux…..’. It has to be water, not water flux.

 It has been improved.

 3.      Pg 2 Line 51 – the disadvantage of using membranes to treat pool water has been mentioned. But in the next paragraphs, the novelty of this work has to be explained. Also, how does the current work overcome the disadvantage may have a good gist of what the paper is all about.

 It has been improved.

 4.      Section 2.3 – Can the authors give a schematic to better understand the circuits involved in membrane filtration using ceramics and polymers?

 A photo of the first installation was added.

 5.      Pg 4 Line 118 – Why have the authors chose different working pressures (0.1 MPa for ceramics and 0.2 MPa for polymer membranes)? In that case will the comparison on flux and fouling be reasonable?

 These pressures were selected on the basis of preliminary tests and observations from previous studies.

Reviewer 2 Report

455574-Membranes review: Title:  Application of ultrafiltration in swimming pool water treatment system

I have finished reviewing the manuscript submitted for publication in Membranes. The overall suggestion I have is that the paper is acceptable for publication after major revision. The manuscript contains new and valuable results and is worthy to be published. The manuscript “Application of ultrafiltration in swimming pool water treatment system shows that UF process cam be a good alternative for the pool water treatment, however, ‘Abstract’ and ‘conclusions’ should be rethink!

The paper is written in quite good English.

My specific comments and questions are as follows:

In Table 2: The MWCO should be given in Da or kDa!

The membranes’ contact angle should be measured and compared, since the used membranes surface characteristics can be different!

In Figure 2: The measuring errors can be shown! How many parallel experiments were carried out?

In Figure 3: The VRR: volume reduction ratios must be given at the end of the experiments.

From figure 4: more details should be explained in the text, because it has/shown very valuable results.

During the UF process with polymer membrane the chlorine present could decrease the membrane material. For this reason the fluxes may increase and the rejection may decrease.  

In this point of view I cannot understand the main conclusion, namely the polymer membrane had better results! ‘Abstract’ and ‘conclusions’ should be rewrite!!!

Author Response

Response to the review comments

Thank you for all the comments. The work has been improved.

Review Report to Reviewer 2

 1.      In Table 2: The MWCO should be given in Da or kDa!

 It has been improved.

 2.      The membranes’ contact angle should be measured and compared, since the used membranes surface characteristics can be different!

 The values of the wetting angle of the tested membranes were added.

 3.      In Figure 2: The measuring errors can be shown! How many parallel experiments were carried out?

 This experiment was repeated four times, and the observed differences in volumetric permeate flux values did not exceed 5%.

 4.      In Figure 3: The VRR: volume reduction ratios must be given at the end of the experiments.

 These values were added to Table 3.

 5.      From figure 4: more details should be explained in the text, because it has/shown very valuable results.

 It has been improved.

 6.      During the UF process with polymer membrane the chlorine present could decrease the membrane material. For this reason, the fluxes may increase and the rejection may decrease.

 This part of the work has been improved.

 7.      In this point of view I cannot understand the main conclusion, namely the polymer membrane had better results! ‘Abstract’ and ‘conclusions’ should be rewrite!!!

 It has been improved.

Round 2

Reviewer 2 Report

I have finished reviewing the revised manuscript and I think that this is more scientific version. I can accept the answers and the changes! My oppinion is that this submission can be acceptable for this Journal!